# Local Chicken Breeds of Africa: Their Description, Uses and Conservation Methods

**DOI:** 10.3390/ani10122257

**Published:** 2020-11-30

**Authors:** Tlou Grace Manyelo, Letlhogonolo Selaledi, Zahra Mohammed Hassan, Monnye Mabelebele

**Affiliations:** 1Department of Agriculture and Animal Health, College of Agriculture and Environmental Sciences, University of South Africa, Florida 1710, South Africa; manyelo.t.g@gmail.com (T.G.M.); letlhogonolo.selaledi@up.ac.za (L.S.); zahrabattal@gmail.com (Z.M.H.); 2Department of Agricultural Economics and Animal Production, University of Limpopo, Sovenga 0727, South Africa; 3Department of Zoology and Entomology, Mammal Research Institute, Faculty of Natural and Agricultural Sciences, University of Pretoria, Hatfield 0028, South Africa

**Keywords:** chickens, conservation, nutrition, breeding, poverty

## Abstract

**Simple Summary:**

In African countries, there has been little effort to conserve the local chicken breeds or lines. These chickens are hardy and provide a valuable protein source to rural households. Even though these chickens are resistant to disease, they are associated with low productivity. However, any improvement in the productivity of local chickens would require close attention to nutritional, breeding and health aspects. Therefore, the main purpose of this review is to provide a detailed understanding on description, uses and conservation methods of local chicken breeds of Africa. In conclusion, this review will highlight information for stakeholders to enable them to make informed decisions on local chicken breed diversification, conservation, and improved production through efficient management, breeding, and nutrition practices.

**Abstract:**

There has been a research gap in the genetic, physiological, and nutritional aspects of indigenous chickens of Africa over the past decade. These chickens are known to be economically, socially, and culturally important to the people of Africa, especially those from marginalised communities. Although they are associated with poor productivity in terms of the number of eggs laid, most consumers prefer their flavoursome meat. Several local chickens have been classified into breeds or ecotypes, but many remain unidentified and are facing extinction. To prevent this, the Food and Agriculture Organization has launched an indigenous poultry conservation programme. In addition, the Agricultural Research Council in South Africa has established a programme to protect four local chicken breeds. The purpose of this review is to provide a detailed understanding of the description, uses and conservation methods of local chicken breeds of Africa. Several studies have been conducted on the nutritional requirements of local chickens, but the results were inconclusive and contradictory. This review concludes that local chickens play a significant role in improving livelihoods, and strategies to preserve and sustain them must be intensified.

## 1. Introduction

Indigenous chickens (*Gallus domesticus*) are chickens that are adapted to harsh environmental conditions that include extensive small-scale village, free-range and organic production systems. Sometimes such chickens are referred to as traditional, scavenging, backyard, village, local or family chickens [1]. In this review, the term *local chickens* is used. The local chicken production system, which is mostly free-range (extensive), can best be described as a low input−low output system. The variations in local chickens mostly comprise plumage colour, body size, feather patterns, comb types and shank colour. In literature, local chicken populations are often described and grouped according to geographical location or phenotypic characteristics, while their classification into breeds or types is limited. Only a few of them have been classified into ecospecies based on their characteristics. 

Throughout the world, numerous indigenous or local chickens have been reported. Naked neck chickens with normal frizzle feathers are reported to be found in Nigeria, Ethiopia, and Southern Africa as explained by Adelake et al. [2]. Whereas large Baladi and Betwil have been reported by Mohammed et al. [3] as local chickens in Sudan. Venda, Koekoek and Ovambo chickens have been described as dominant local chickens with Ovambo originating from northern part of Namibia [4]. According to Alewi et al. [5] the local *Kei* (a red plumage chicken) in Ethiopia is known as the main local chicken breed. Local or indigenous chickens are more abundant mostly in developing and underdeveloped countries than those that are already developed.

Numerous studies have shown that local chickens play a key role in improving the socio-economic status of many rural communities. However, poor housing, lack of coordinated disease control mechanisms, poor feeding and the absence of conservation strategies are some of the challenges facing local chicken production systems in Africa [6]. Parasitism in the intestines of local chickens is another problem and results in low weight gain and poor carcass quality [7]. Distance to the nearest market, access to extension services, feed costs, market price and the education level and experience of farmers are further factors that can affect the profitability of local chicken rearing [8,9]. Despite these challenges, local chickens are a source of income and protein to resource-limited local marginal communities in developing countries. Local chickens are preferred over exotic chicken breeds because of their succulent meat. They also sell at a cheaper price [10,11]. Hence, the demand for local chicken products (eggs and meat) is high. It is estimated that local chickens constitute 80% of poultry production in sub-Saharan countries [12], with Nigeria known to have the highest number of local chickens with an estimated population of 180 million [13]. Though these figures demonstrate the necessity to increase the production of poultry [14], the quantity and quality of the product are yet to improve [10].

The flock size of and mortality rate among local chickens in African countries vary. In Southern Africa, different studies have reported mean flock sizes of 12.9 to 29.98 chickens per household [15,16], while mortality among local chickens in, for example, Namibia is estimated at 42.2%, which is the highest mortality rate in southern African countries [17]. Several reasons have been put forward for this, which include mismanagement, malnutrition, diseases, and predation [16]. Thus, flock size is mainly affected by predation, diseases, and theft [18]. As for their production quality, local chickens can lay 20 to 80 eggs per year, which is very low compared with commercial breeds that can lay up to 300 eggs per year. Nutritional deficiency and low genetic potential are some of the factors influencing the low production of eggs [13]. Hence, genetic material must be improved to enhance productive efficiency [19,20]. 

Overall, local chicken farming in southern African countries remains at a developing stage [10]. A case in point is Zambia, where only 0.5% of the total local chicken population reaches the commercial market, with the majority being consumed within a household [21,22]. Thus, regardless of their importance, local chickens have received little attention in terms of improving their production rates [23]. Many researchers from African countries have addressed the challenges related to improved nutritional management and genetic upgradation of local chicken, but there is limited information on how improvement at these levels can enhance performance [18]. According to a study conducted by Badubi et al. [24], the local chickens of Botswana are considered to be bigger than the local chickens of other African states. Nthimo et al. [25] conducted another comparison study and found that Lesotho’s local chickens are the poorest performers in all production traits compared with other southern African indigenous bloodlines. The factors that contribute to this poor performance are complex, but their nutritional and genetic development appear to be areas that can be explored and should, therefore, be a priority. Therefore, conservation decision-making should look at traits of scientific or economic importance, adaptation to a specific environment, the historical or cultural importance of the species and the degree of extinction [26]. Whether the process of conservation should be carried out within (in-situ) or away (ex-situ) from natural habitats would depend on the conservation objectives. Therefore, the objective of this review is to collate current information on the description of indigenous chickens of Africa and current conservation strategies with a view to highlight improvement at their nutritional and genetic levels of performance.

## 2. Methodology

This review was conducted according to the guidelines in the Preferred Reporting Items for Systematic Reviews and Meta-Analyses (PRISMA) statement [27]. A comprehensive search was conducted to identify eligible studies using a five-stage process. In the first stage, a search to obtain all relevant studies that were published before August 2020 was performed using databases such as the Web of Science, Science Direct, Google Scholar and the Wiley Online Database. The search strategy involved a combination of the following keywords: indigenous chickens, utilisation, conservation, nutritional requirement, breeding, and genetics. The search was not restricted by language, date, or study type and 95,100 studies were listed. During the second stage, the search was narrowed down by adding the words “southern Africa”. This resulted in 51,900 references. Furthermore, the search was narrowed down to the time scale of 1999 to 2020 (the period was chosen to capture as wide a range of articles as possible) and 33,800 results based on the title and keywords were obtained and examined. In the third stage, 5833 studies were eliminated for differing reasons. The exclusion criteria included articles where the abstract could not be found and written in a language that could not be understood by the authors (i.e., German, Dutch, Spanish or Italian). A final total of 108 remaining full-text studies on indigenous chickens were consequently assessed for eligibility. The fourth stage involved the reading of article titles and abstracts through screening of the retrieved articles. Thereafter, the full-length individual manuscripts were screened and papers not satisfying the inclusion criteria were excluded. In the fifth stage, the remaining additional literature was included through the examination of the reference lists in the literature extracted, academic resources (master’s and doctoral dissertations), PLoS ONE and the Directory of Open Access Journals.

## 3. Description of Local Chickens of Africa

It is believed that the local chickens of Africa originated from South-East Asia, China, and India [28,29]. Local chickens are hardy and can adapt to local conditions better than other breeds because of their ability to withstand harsh climatic conditions due to their typical genetic development [30]. Further to this, they also possess a strong ability to anticipate danger and act quickly. Alders and Spradbrow [31] stated that local chickens have a strong ability to fly and run to escape dangers and predators in comparison with commercial chickens. These traits make them strong enough to survive and produce in an unfavourable environment. Mammo et al. [32] indicated that the husbandry of local poultry is, to a large extent, associated with resource-poor farmers who keep the chickens under extensive systems. Furthermore, Nhleko et al. [33] summarised the characters of village chickens, stating that they are among the most adaptable domestic species with an ability to survive in cold or warm and wet or dry conditions, sheltered in cages or unsheltered outside or roosting on tree tops. Van Marle-Köster et al. [30] stated that the fowl found in the rural areas of Southern Africa are mostly named and classified based on their phenotype and geographical location. 

However, most rural communities do not commercialise free-range chicken production. According to Mtileni et al. [6], local chickens are mostly raised as part of mixed farming in extensive systems and, to a lesser degree, in semi-intensive systems. Backyard local chickens provide rural communities with a means of converting available feed resources around the household or village into highly nutritious products such as meat and eggs [28]. Moreover, Swatson et al. [34] stated that the reason for most poor rural communities to keep local chickens is for religious purposes, food security, and socio-economic and cultural considerations. Farmers in these areas consider chickens as their primary source of domestic animal protein. A study conducted in Mozambique by Harun and Massango [35] showed that village poultry production has the capacity to improve food security, assist in poverty reduction and mitigate the adverse economic impacts of Human Immunodeficiency Virus/Acquired Immunodeficiency Syndrome (HIV/AIDS) on the locals. 

According to Swatson et al. [34], a complex interaction of biological, socio-economic, cultural, and agro-technical factors is the reason why these communities do not commercialise free-range chicken production. For example, lack of household training in poultry management, veterinary support, feeding practices and the use of improved indigenous breeds play a role in rural farmers failing to practise commercial farming [6,35]. Thus, in poor rural communities, it is necessary to ensure the sustainability of free-ranging indigenous poultry development projects.

Local or indigenous chickens have the highest rate of variation of population types among chicken species [26]. In African countries, such as Nigeria, Zimbabwe, Namibia, Kenya, Malawi, Sudan, Ethiopia, Southern Africa etc., local chickens are characterised by a great variation observed in morphological characteristics and production parameters [36]. Most African local chickens have a distinguished plumage pigmentation whereby some tend to have blackish and brownish colours showing extended and pied colourations (Table 1) with normal plumage distribution whereas some chickens have special forms such as naked neck frizzle and silkiness [26]. Regarding body weight, some tend to be dwarfs, heavy or normal. Mostly, the comb of local chickens is single with crest present [22]. Normally, differences on morphological characteristics of local chickens are due to different climatic conditions. According to Apuno et al. [37] local chickens from hot climatic conditions are characterised by large single comb, naked neck and frizzle feathers, which allow efficient heat regulation. Whereas local chickens from cold climates are characterized by a lot of feathers covering their body and this helps with insulation and protection against losing body heat [38]. Most African countries consist of the following indigenous chickens breeds: the Naked Neck, the Ovambo chicken, the Potchefstroom Koekoek, and the Boschveld and Venda chickens. Mosoeunyane and Nkebenyane [39] stated that around 84% of the households in Lesotho’s rural areas depend on native chicken production for food security and income generation. Each breed is described in detail below.

### 3.1. Naked Neck Chickens

The African Naked Neck is thought to have originated in Malaysia [26] and two types exist. The first is considered purebred with a complete naked neck (Figure 1), while the second is regarded as not purebred with the front part of the neck having a tassel. Indigenous chickens with total naked necks are the result of mating between two tasselled birds [29]. Naked Neck chickens have single red combs with a large wattle. They are characterised by reddish bay eyes and red earlobes. The average weight of roosters and hens ranges between 1.5–3.5 and 1.1–3 kg, respectively [39]. The Naked Neck breed can produce up to 138.9 eggs annually, while the hens tend to produce their first egg at 129 days [47]. 

Venda chickens are a distinctive multicoloured breed with white, black, and red as the dominant colours (Figure 2). These breeds are named after their origin, and they were discovered by veterinarian Dr Naas Coetzee while he was doing research in Venda in the Limpopo province, South Africa [32], in 1979. Later on, similar chickens were seen in the southern Cape and in Qwaqwa [48]. These chickens are large and lay large, tinted eggs. The average weight of the cockerels and hens can reach up to 2.9–3.6 kg and 2.4–3.0 kg, respectively. Hens of the Venda breed produce large, slightly pink-tinted eggs with the average egg weight of 53 g. They reach sexual maturity at around five months of age.

### 3.2. Ovambo Chickens 

Ovambo chickens originate from the northern part of Namibia and Ovamboland in Africa. The Ovambo breed is dark-coloured and smaller in size (Figure 3). It is known to be very aggressive and agile due to its habit of catching and eating mice and young rats. These chickens can fly and avoid predators by climbing to the top of trees [41]. Ovambo chickens are a dual-purpose breed [49]. Sexual maturity is attained at 143 days and males can reach a weight of up to 2.16 kg, while females can weigh up to 1.54 kg. 

### 3.3. Potchefstroom Koekoek 

The Potchefstroom Koekoek is a cross breed between the White Leghorn, the Black Australorp, and the Barred Plymouth Rock [50]. Its name is derived from its colour patterns (Figure 4) [49]. It is a dual-purpose breed that lays brown-shelled eggs with an average weight of 55.7 g [4]. It has a sex-specific feather colour and pattern and reaches early sexual maturity at 130 days. The average weight is around 3–4 kg for males and 2.1 kg for females [43]. The Potchefstroom Koekoek reaches sexual maturity at 138.5 days [41]. The hens are known for being broody and good sitters. The hatchability rate reaches up to 78%.

### 3.4. Boschveld Chicken 

Boschveld chickens are recognised as a synthetic indigenous chicken breed (Figure 5). They were bred at the Mantsole ranch in the Limpopo province of South Africa in 1998 from a cross between three indigenous African breeds (Venda, Ovambo and Matabele chickens). They were bred for the production of eggs and meat under harsh African conditions. It is stated that the Boschveld chicken inherited 50% of the traits of the Venda breed, 25% of that of the Ovambo breed and 25% of the traits of the Matabele breed [44]. The Venda breed is fairly large and lays large, tinted eggs. It is known for its good mothering ability and weighs roughly 2.5–3 kg. Not much additional information is published on the breed [45]. However, its offspring has shown high egg production ability and good meat quality. It has a medium body frame (average weight at 20 weeks between 1.7 and 2.6 kg) and the ability to fend on its own [52].

To date, local chickens, including the Naked Neck, Venda, Ovambo, Potchefstroom Koekoek and Boschveld breeds, appear to be the most upcoming ecotypes in Southern Africa. The performance and productivity of these ecotypes have been studied and documented. Furthermore, their importance in solving many of the socio-economic problems most rural communities are facing has been proven [21].

## 4. Importance of Local Chickens in Alleviating Household Poverty

Despite the efforts of different stakeholders, attainment of food security remains a challenge in many African countries. As mentioned, local chickens play an important role in providing a cheaper source of quality protein for local communities [53]. Moreover, although local chickens produce less meat and fewer eggs in comparison with conventional chickens, they have an important role to play in providing food security and a source of income generation to resource-limited local communities who rely on them at a socio-economic level [54]. At a social level, the meat of local chickens is preferred by many because of its unique and succulent taste [55], as mentioned earlier. Apart from being kept for food, communities in the southern African region also kept local chickens for cultural and religious purposes, e.g., white and black magic and cock fighting, in the past [56]. In Zimbabwe, local chicken meat is reserved for special guests or served at ceremonies such as marriage, weddings or funerals [57], and some farmers present the chickens as gifts to friends and families as tokens of appreciation [58]. 

At an economic level, local farmers depend on chicken sales as a source of income. The advantage in this regard is that these chickens are mostly reared with limited outlay, which makes them affordable for almost every household. A study by Gunya et al. [59] reported on the contribution of local chickens towards income generation for local farmers. In South Africa, an average of R80 (USD 7.22) per chicken has been found to be the amount a farmer can generate from the sale of chickens [59]. However, it is evident that the low production rate among local chickens negatively affects the decision of whether to sell or to consume. McAinsh et al. [60] found that, in Zimbabwe, women own most of the chicken flocks and that the income from chicken production is mostly spent on improving the nutrition, health and education of their households. In some cases, local chickens are used as “cushions” or “banks”. As such, they are “guarded” to be sold in hard times to pay for school fees, medical costs and other uncertainties [61]. However, according to a report by Mtileni et al. [6], only a small portion of the population in Venda uses local chickens to generate income despite a good number of households owning village chickens. Furthermore, Muchadeyi et al. [57] stated that, while it is evident that local chickens contribute towards food security and income generation among the rural communities in Zimbabwe, their monetary contribution to the Zimbabwean economy is still viewed as low. Kusina and Kusina [58] attributed some of the challenges that are related to this to the lack of markets and marketing skills among the rural communities. The training of local farmers in marketing skills and interventions to improve infrastructure for chicken transportation to places of sale could improve the socio-economic status of farmers. Therefore, it is of paramount importance that these endangered local chicken breeds are preserved in order for them to continue to contribute to poverty alleviation in marginalised communities.

## 5. Conservation Strategies to Preserve Local Chicken Breeds

It is believed that 33% of local chicken breeds are facing extinction [61,62]. Although Mahrous et al. [63] reported that frizzle-feathered and Naked Neck chickens have adapted well in tropical temperatures above 25 °C, they remain among breeds such as dwarf chickens and Silkies that are reported to be seriously endangered [64]. This problem is not a challenge in southern African countries alone [60,61]. It is a problem facing developing countries in general and must be addressed through the implementation of conservation programmes. Consequently, the conservation of indigenous chickens is currently receiving much-needed attention in sustainable animal breeding segments simply because of the distinctive genetic resources of these breeds [65], the conservation of which is of critical importance. There are many conservation decision-making frameworks that can be implemented. However, the most commonly used are structured decision-making, systematic conservation prioritisation and systematic reviews [66]. These involve activities such as developing strategies and management programmes, planning, designing policies and implementation. An important and essential strategy that can be adopted in conservation decision-making processes is the genetic characterisation of indigenous chicken populations [62]. This subject can be addressed fully by implementing knowledge-based animal genetic resources management [67].

Mtileni et al. [68] found a high degree of heterozygosity and a number of alleles in local chicken populations in South Africa and Zimbabwe. Therefore, a sustainable method of conservation is through matching the different genotypes to the appropriate environment [25]. However, because of the already-mentioned inadequate characterisation of local chickens in most southern African countries, it is difficult to fully understand the present diversity that could be used to formulate and implement conservation strategies. Creating an inventory of local chickens is a fundamental step towards breed conservation and future breeding strategies [67].

The Food and Agriculture Organization (FAO) of the United Nations has initiated a programme for the characterisation and conservation of indigenous livestock breeds in the hope of conserving the genetic material of these breeds [62]. The South African Agricultural Research Council (ARC) also has a programme for the genetic improvement and conservation of four local breeds, namely the Naked Neck, the Ovambo, the Potchefstroom Koekoek and the Venda [68]. However, the situation might be different in other southern African countries due to limited resources [57]. 

Conservation strategies are generally grouped into in-situ and ex-situ strategies [61]. In-situ conservation is characterised by traditional production systems where breeds are maintained in their environment or on-farm, while ex-situ conservation involves maintaining breeds outside their traditional production systems using technological advances such as cryopreservation [69]. 

The majority of poultry genetic resources are preserved in-situ in the living population. However, this process has its challenges such as pathogen epidemics, genetic problems, or natural calamities [70]. Ex-situ conservation of chickens is mainly practised by industrial chicken farms through the collection of frozen semen. However, indigenous breeds are fully preserved by way of in-situ populations [70].

An in-situ conservation programme for indigenous chickens will involve different stakeholders. Governments are the regulatory bodies responsible for providing mechanisms to determine the “how” and “who to be involved in” of conservation programmes. Other stakeholders include breeding centres as the nuclei of pure breeding stock, farmer groups as the backup breeders of pure breeds, farmers as the central figures on whose interests the conservation programmes should be based and, lastly, research institutions or universities as important reservoirs of indigenous chickens and sources of capacity development among farmers [71]. Ex-situ conservation programmes may have some limitations in developing countries. However, their main focuses are cryopreservation and reproduction technologies for the conservation of genetic resources. These are biotechnology tools that create gene banks as physical repositories where samples of a genetic resource (semen, embryos, live animals, oocytes, tissues, and DNA) are preserved. This can be supported by data banks where information related to the characteristics of a species is stored in a systematic manner [70]. The technological advancement of primordial germ cells transplantation provides insight into ex-situ conservation as it enables the capture of the entire genetics of the stock [72]. Thus, conservation strategies can be implemented with the help of southern African government regulatory bodies to prevent the extinction of local chicken breeds.

## 6. Improvement of the Performance of Local Chickens through Breeding and Genetics 

The breeding and genetic improvement of local chickens are complex processes but essential components of agricultural sustainability. They are based on the selection of breeds to be used and involve a good understanding of the specific attributes or phenotype of each breed [57]. Animal breeders have been employing these techniques to breed more resilient and productive agricultural animals. Innovations such as artificial insemination and semen collection and preservation have also long been applied in the food and animal industries. However, many factors must be considered when planning genetic improvement in local chickens. For example, tolerance to high temperature as a key requisite [73] can be improved by incorporating single genes that modify or reduce feathering such as naked-neck, scaleless and frizzle genes [74]. Breeding strategies such as crossbreeding, upgrading through back-crossing and within-line selection can also be used to improve local chickens [63]. Especially crossbreeding can be used as an essential mechanism to improve the growth performance and bodyweight of local chickens raised under an intensive management system [75]. However, there is limited scientific information on the crossbreeding of local chickens to improve growth traits in Southern Africa [76].

Furthermore, the genetic improvement of chickens in developing countries has focused on exotic breeds [77]. The lack of productivity among local chickens in Zimbabwe, for example, can be ascribed to a lack of genetic improvement of the qualities that are considered to be commercially viable [70]. Inbreeding can also reduce the productivity of local chickens and should be avoided. However, literature on the genetic improvement of southern African local chickens is scare. There is, therefore, a need to develop and implement coordinated genetic improvement strategies aimed at local chickens. Table 2 describes the crossbreeding strategies used in different southern African countries to improve the productive performance of local chickens.

## 7. Enhancement of the Performance of Local Chickens through Nutritional Strategies

Providing the right nutrition is important for the growth, production, and health of poultry. Furthermore, depending on factors such as bird age and production status, different nutrients are required. This also applies to local chickens to ensure that they achieve their productive potential and remain healthy. Poor-quality feeds and incorrect mixing of dietary nutrient levels such as energy and protein can, potentially, cause nutritional stress and health concerns among local chickens [81]. Chickens vary greatly according to the purpose they have been developed for. Those raised for egg production have a small body size and are known as layers, whereas those raised for meat are known as broilers. The feeding methods for these two kinds of chickens differ, since they are not grown for the same purpose. According to Tang et al. [82] and Mohammad and Sohail [83], efficient nutrition can be used is an important factor in determining the performance and productivity of chickens and there is evidence that the productivity of broiler chickens can be improved by the manipulation of their diets. Thus, to be competitive, the poultry industry in this region will have to continue improving the productivity of local chickens through more efficient nutrition. Although improvement of local chicken breeds in the rural areas of Southern Africa through efficient nutrition has been slow, the development and formulation of diets that match the nutrient requirements of these breeds are now growing rapidly. 

Several studies have been conducted to improve the productivity of local chickens through efficient feeding (Table 3). According to Alabi [84], protein, lysine (a base for all other amino acids) and energy requirements of local chickens must be met optimally in order to improve and maximise productivity. According to Kingori et al. [36], a protein level of 16% in local chickens aged between 14 and 21 weeks has been observed to optimise feed intake and growth. However, protein levels of 17% to 23% have been reported as not to have any effect on the growth and feed intake of local chickens [85]. On the other hand, Mbajiorgu [86] reported that 18% to 19% crude protein content levels showed improved growth and productivity in indigenous Venda chickens.

Several authors have also studied the energy requirements of local chickens [78,80,81]. Energy level is usually selected as the starting point in formulating poultry diets. An appropriate energy level in the diet is one that will most likely result in the lowest feed cost per unit of product. According to Nahashon et al. [89], the control of feed intake is believed to be primarily based on the amount of energy in the diet, which results in chickens consuming a lot of food to meet their energy requirements. Mbajiorgu [86] reported that indigenous chickens aged between one and six weeks old require a diet containing an energy level of 14 MJ ME per kg Dry Matter (DM) for optimal growth. According to Alabi et al. [84], a diet containing 12.34 MJ ME/kg DM to 12.91 MJ ME/kg DM is recommended for optimal growth performance during the starter and grower phase of Venda chicks under well-controlled management conditions.

Chicks tend to lose weight immediately after hatching and this has adverse effects on their subsequent productivity [90]. This may be related to nutritional limitations immediately after hatching, such as the adaptation of the gut to solid feed [91]. Hence, several studies have looked at nutritional strategies to improve the productivity of chicks [92,93]. According to Ng’ambi et al. [94], egg weight affects hatchability, chick hatch-weight, mortality and the subsequent productivity of local chicks fed a grower diet containing 880 g DM/kg, 16.8 MJ energy/kg DM, 200 g crude protein/kg DM, 11.5 g lysine/kg DM, and 25 g fat/kg DM. Alabi et al. [95] also reported an increase in the weight of Venda and Potchefstroom Koekoek chicks hatched from heavy eggs from chickens fed different dietary energy levels (11 MJ ME/kg DM, 12 MJ ME/kg DM, 13 MJ ME/kg DM and 14 MJ ME/kg DM). Moreover, authors have reported higher mortality rates among Venda chicks hatched from heavier eggs [94,95].

A number of authors have also highlighted deficiencies in the diets of local chickens that would require attention to formulate strategies to overcome nutritional problems [96]. According to Van Ryssen et al. [97], wood ash is a good source of calcium that can be used in the diets of local chickens as a supplement to replace feed lime. It is well documented that soil ingestion by animals can have a significant effect on trace mineral ingestion, implying that elements such as iron in soil are bio-available [97]. Geerligs et al. [98] suggested that food prepared in iron cooking pots could be used to overcome Fe deficiency in developing countries. It would, therefore, appear that local chickens do not need mineral supplementation to overcome nutritional problems since they can access it easily. Protein and energy, however, are the main nutrients that need a lot of attention in respect of nutritional problems among local chickens.

## 8. Poultry Welfare, Housing, Diseases and Predators

Feed resources, housing diseases, parasites and predators are challenging issues that local poultry keepers face. Understanding the husbandry practices can assist greatly in the management of rearing local chickens. Local chicken production systems are normally practiced under free range systems and the major proportion of the feed is obtained through scavenging [13]. Normally these chickens are reported to depend more on insects, worms, and plant materials, with very small amounts of seeds and table leftover supplements from their keepers [99,100]. Studies have shown that there is no purposeful feeding of rural household chickens and the scavenging feed resource is almost the only source of feed [14]. However, the type and amount of feed given to local chickens normally depend on the crops grown in the area as well as the seasons. Most local chicken farmers practice supplementary feeding systems (mostly once per day) by giving their chickens maize, barley, wheat, finger millet and household waste products to feed them. After hatching, the chicks could forage and roam freely with their mothers in open areas near the home and surroundings.

### 8.1. Housing 

The rearing of local chickens requires simple backyard housing systems mostly built with cheap and readily available materials. This can be achieved by not separating the chickens’ house from the yard of the family that own them [101]. According to Mearg [102], lack of housing is one of the constraints of raising local chickens. Normally, high mortality in local chicken breeds is due to predators because of improper housing [103,104]. Several studies have reported that there is no special housing provided for local chickens. Most of the time, poultry houses are made of two or three raised parallel planks of wood or eucalyptus poles and branches [105]. Some use hand-woven baskets to shelter their chickens at night [102]. However, a study by Mengesha et al. [11] reported that smallholder poultry farmers share the same house with the chickens but provide separate room for them.

### 8.2. Diseases, Parasites and Predators 

The rearing system of local chickens is known to have high mortality caused by predators, parasites, and Newcastle disease, but it can be reduced by improved management and basic disease control, including vaccination [105,106]. However, parasites and diseases are the chief hindrance in raising local chickens [107]. Protozoa and worms are the most common types of internal parasites that normally affect poultry. Usually, low levels of infestation of these parasites do not cause a problem and can be left untreated. Normally, poor growth and feed conversion ratio and decreased egg production are clinical signs of a parasite infestation in poultry. Moreover, parasites can make a flock more susceptible to diseases and in most cases death may occur. 

Chickens ingest these parasites through contaminated water and feeds. There is paucity of information on monitoring parasite and disease infection in commercial and rural farms of South Africa. Any pest and disease challenges, such as Newcastle disease (NCD) and avian influenza virus, can have devastating effects on the economy and food security of the country. Newcastle disease is one of the major and economically important constraints for local chicken production systems [101,108]. Predators are also known to contribute to local chickens’ mortality alongside diseases as a major cause of premature death. Normally, prey such as vultures and wild birds (eagle, hawk, etc.), which only prey on chickens and wild mammals such as cats and foxes, are well known predators that prey on mature birds as well as chicks [10,106]. Local chickens can be protected from all of these by proper health management practices and proper housing. 

## 9. Conclusions

This review has provided an insight into the importance of local chicken breeds in improving and maintaining livelihoods in Africa. Few chickens have been identified into breeds, with very limited conservation programmes available in these regions. In addition, very few studies have been conducted on the nutrient requirements of these breeds to date. Such requirements, therefore, remain inconclusive. Several studies have been performed on the crossbreeding of local chicken ecotypes, and some have yielded superior breeds such as the Boschveld chicken. It is important to note that more work needs to be undertaken, and it should be government policy to encourage the conservation of these breeds in order to avoid extinction. Therefore, conservation strategies and genetic improvement can be developed and implemented simultaneously in a coordinated approach. This strategy will allow new progeny to be developed and studied while the original breed is preserved.

## Figures and Tables

**Figure 1 animals-10-02257-f001:**
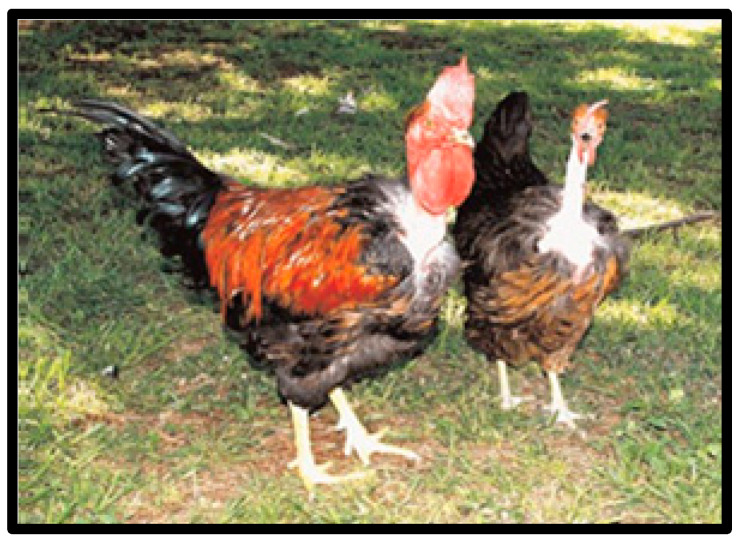
Naked Neck chicken breeds. Source: [42].

**Figure 2 animals-10-02257-f002:**
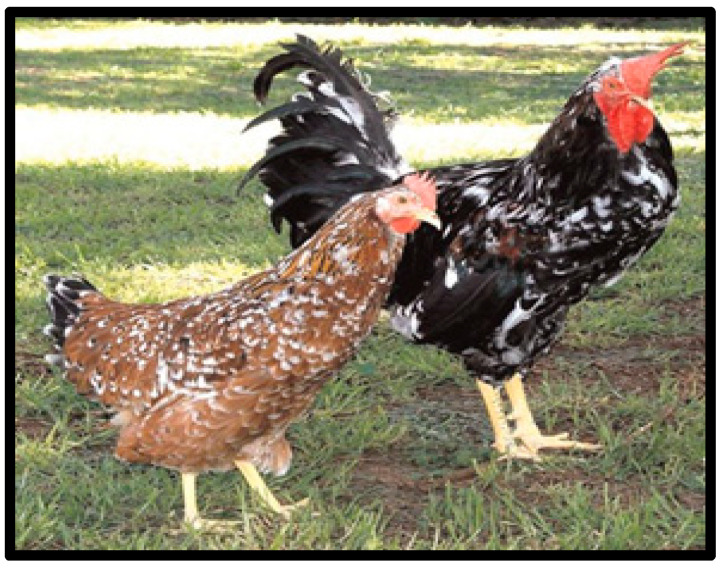
Venda chicken breeds. Source: [42].

**Figure 3 animals-10-02257-f003:**
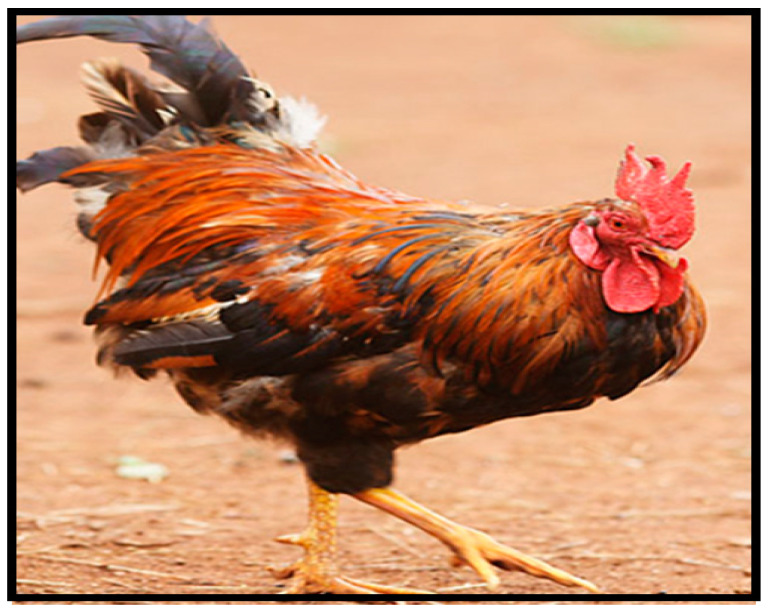
Ovambo chicken breed. Source: [4].

**Figure 4 animals-10-02257-f004:**
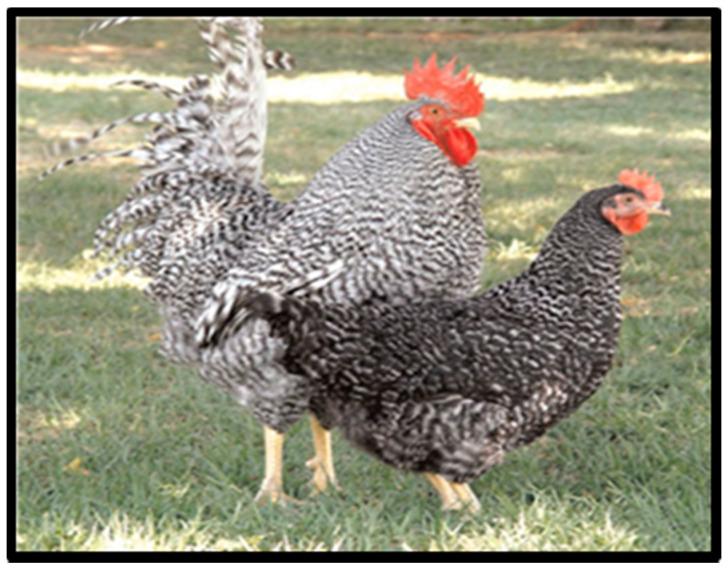
Potchefstroom Koekoek chicken breeds. Source: [51].

**Figure 5 animals-10-02257-f005:**
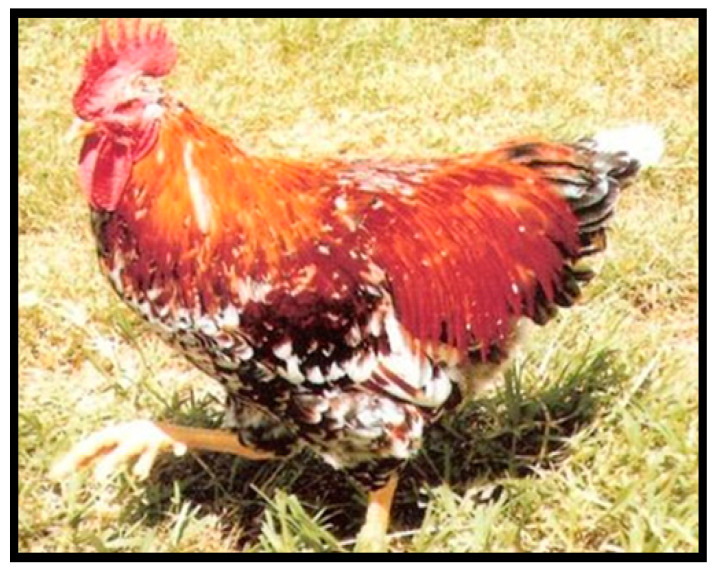
Boschveld chicken breed. Source: [42].

**Table 1 animals-10-02257-t001:** Summary of distribution, phenotypic characteristics, production/performance traits of local/cross breed chickens.

Breed	Distribution	Phenotypic Characteristics	Production Data	Researchers
Naked Neck	Introduced to Africa by traders from Malaysia	Very colourful Naked Neck major gene, plain head and single comb, with medium-size wattles in females and highly-developed wattles in males	Increased weight gain and dressing percentage, adaptation to high environmental temperatures has superior egg production and egg quality and resistance against disease	[32,40]
Venda	Southern Africa	White and black/white & brown plumage, green on feather tips	Quality of egg production, self-sustainment, resistance against diseases, low need for food and broodiness	[4,32]
Ovambo	Northern part of Namibia and Ovamboland	Brown & black plumage, aggressive birds	High egg and meat production	[4,33,41]
Potchefstroom Koekoek	Southern Africa, Potchefstroom Research Station 1950’s	Its black and white striped feathers, with distinct patterns in the roosters and hens.	Lays brown-shelled eggs with an average weight of 55.7 g hatchability rate reaches up to 78%.	[32,42,43]
Boschveld	Mantsole ranch in the Limpopo province of South Africa in 1998	Light brown and white feathers	High egg production ability and good meat quality	[40,44,45]
Tswana	Botswana	Vary with colours: black, frizzle feathers, typical brown mixed with white colour	Low egg production, egg size range between 38–60 g	[46]

**Table 2 animals-10-02257-t002:** Improvement of performance or production of local chickens at a genetic level.

Breeding Programme/Strategy	Indigenous Chickens Used in Southern Africa	Improved Production Parameter	Researchers
Crossbreeding	Potchefstroom Koekoek, Venda, Ovambo and Boschveld	Crosses between Venda and Potchefstroom Koekoek as well as Ovambo and Venda provided the highest heterosis effect on body weight	[76]
Crossbreeding	Black Australorp × Tswana crossbred	Body weight was higher in Australorp × Tswana crossbred males and females than their indigenous purebred counterparts	[78]
Crossbreeding	Black Australorp × indigenous naked neck Tswana chickens	Body weight was higher in Australorp × Tswana crossbred males than purebred males from 10 weeks. Crossbred females were heavier than their purebred counterparts	[75]
Crossbreeding	Crossbred chickens and purebred Tswana chickens	Body weight was higher in crossbred males and females than purebred Tswana chickens. Introducing the Orpington breed to the Australorp × Tswana crosses did not boost growth performance	[79]
Crossbreeding	Commercial Rhode Island Red (RIR), local and crossbred chickens. Crossing RIR layer cocks to local hens	RIR and crossbred chickens had higher egg weight, egg length, egg breadth, egg volume and chick hatch weight than local chickens	[80]

**Table 3 animals-10-02257-t003:** Recommended protein and energy for improved productivity and performance among local chickens.

Nutrient Level	Improved Production Parameter	Indigenous Chickens Age	Researchers
Protein			
16%	Feed intake per bird increased with increasing dietary protein level	Between 14 and 21 weeks	[36]
17 to 23%	Had similar growth rates and feed intakes	1 and 6 weeks	[85]
18–19%	Improved growth and productivity	Between one and six weeks	[86]
18%	Improved weight gain	Between one and 13 weeks	[87]
15.53%	Improved carcass weight and percentage	14 weeks	[88]
Energy			
14 MJ ME/kg	Improved growth and productivity	Between one and six weeks	[86]
12.34 MJ ME/kg	Improved feed intake, growth rate and feed conversion ratios	Between one and seven weeks	[84]
12.91 MJ ME/kg	Improved feed intake, growth rate and feed conversion ratios	Between eight and 13 weeks	[84]
2750 kcal/kg ME	Improved growth performance	Between nine and 20 weeks	[83]
2842–3200 kcal/kg ME	No significant difference in growth performance parameters	Between six and 9 weeks	[85]

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
