# Peer review of "Local Chicken Breeds of Africa: Their Description, Uses and Conservation Methods"

_animals, 2020, doi:10.3390/ani10122257_

Round 1

Reviewer 1 Report

I thank the authors for doing a thorough revision of the manuscript. All the suggestions/corrections have been accepted and queries were duly answered. The manuscript looks very improved now and I recommend its publication.

Author Response

Dear Reviewer,

Authors would like to thank and acknowledge comments and suggestions made to improve manuscripts.

Regards,

Reviewer 2 Report

The authors have addressed my first comment about the need to include health and welfare by adding a section.

However, the authors have not addressed my second comment that this review is relevant to their local conditions only, and is not appropriate for an International journal. For example:

(Line 18) "Therefore, the main purpose of this review is to provide a detailed understanding on description, uses and conservation methods of local chicken breeds of southern Africa"

(and in the conclusion, line 436) "This review has provided an insight into the importance of local chicken breeds in improving and maintaining livelihoods in southern Africa"

The purpose of the manuscript and conclusions, as the authors state, relate to the local conditions, and would perhaps be more suitable for a regional Journal.

Author Response

Dear Reviewer,

Authors would like to thank and acknowledge comments and suggestions made. Manuscript has been improved and provided an insight into the importance of local chicken breeds in improving and maintaining livelihoods in other African countries to be appropriate for an International journal like Animals.

Regards,

Reviewer 3 Report

The authors have made numerous modifications according to the modification comments, but there are still some problems to be further improved:

  1. Commas are missing in some sentences. For example, in line 16, the sentence should be changed to “Even though these chickens are resistant to disease, they are associated with low productivity.” The same problems also exist in line 29.
  2. Please delete “though” or “but” in the sentence in line 28.
  3. Please add a conjunction to connect two sentences. For example, in line 33, the sentence should be changed to “Several studies had been conducted on the nutritional requirements of local chickens, but the results were inconclusive and contradictory.” The same problems also exist in line 35 and line 80.
  4. Please check if the word “one” is an extra word in the sentence in line 401.

In a word, I think the revised manuscript can be accepted if the above questions have been modified.

Author Response

Dear Reviewer,

Authors would like to thank and acknowledge comments and suggestions made to improve manuscript.

Reviewer 4 Report

The manuscript “Local chicken of Southern Africa: Their Description, Uses and Conservation Methods” is a review about the breeds, conservations programs, nutional strategies, housing, diseases and predators in indigenous chicken. My main criticism is that although the review is ambitious, the authors do not delve into most of the aspects studied.

The title indicates that the revision is made on Southern Africa, but a small number of countries are highlighted. The introduction paragraphs are disjointed and there is no common thread that leads to the objective of the work. In the body of the review, concrete data is missing and in turn many common statements are found without being accompanied by concrete data (one of the objectives of a correct review).

In my opinion, the methodology section is not necessary. A correct revision implies the study of all the relevant works related to the subject.

It is also striking that only 5 local breeds are in this vast territory (Table 1). However, table 2 shows local breeds for crossing that are not in Table 1 (Tswana?).Data on the effective size of populations, breeding areas, productive data, etc. are missing.

I want to emphasize that although the authors have made a great effort to compile this information, which is presumably scarce in many cases, in my opinion, the quality of the presentation makes it not acceptable for publication in a journal of the quality of Animals.

Author Response

Dear Reviewer,

Authors would like to thank and acknowledge comments and suggestions made to improve manuscript.

This manuscript is a resubmission of an earlier submission. The following is a list of the peer review reports and author responses from that submission.

Round 1

Reviewer 1 Report

Please find comments in the attached document.

Author Response

Dear Reviewer,

Authors would like to thank and acknowledge comments and suggestions made. Manuscript was revised and comments and suggestions were addressed accordingly.

Regards.

Reviewer 2 Report

Local Chicken Breeds of Southern Africa: Their Description, Uses and Conservation Methods

I thank the authors for undertaking a review of an important topic. I agree that “local” (back-yard) chickens are important to many communities, and present unique conservation, social, welfare and health (both for chickens and zoonosis) implications that in my opinion warrant more attention.

Despite my enthusiasm for the topic, I’m afraid I can’t recommend publication of this review in Animals. My reasons are:

  • A review article in Animals may be expected to cover the topic with more breadth. The health of local chickens is an important topic that readers may expect to find in a review, but is not covered (A good review on this topic in Sandilands, V., & Hocking, P. M. (Eds.). (2012). Alternative systems for poultry: Health, welfare and productivity (Vol. 30). Cabi.). Similarly, variation in management is important for welfare, yet at the moment management is too narrowly confined to breeding and nutrition. Behaviour, welfare and protection for predators for example are also not covered, but are important in these systems. Rural programmes, information and technologies may also be important in production or contributing to the economy and incomes. In short I think a review article on this topic for this journal should be broader than this article it currently is.
  • Animals in an international journal with an international readership, and the issues described are perhaps of interest mainly to local communities, and less so to international readers. I do acknowledge that the authors have in places tried to consider the wider implications of their findings, and this is something that could be extended. For example, you may present the “Southern Africa” situation as a “case study” that can be used to understand impacts and changes in other parts of the world where local breeds are equally important.

I hope the above suggestions encourage the authors to continue this important work and to strive to disseminate their findings.

Author Response

Dear Reviewer,

Authors would like to thank and acknowledge comments and suggestions made. Health, welfare and protection from predators section has been covered as suggested.

Thank you.

Reviewer 3 Report

This review presented the description, uses and conservation methods of local chicken breeds in Southern Africa. I appreciate greatly the time and effort required to conduct studies such as these, especially on heritage poultry breeds. However, there were some issues I want to discuss with authors, and the paper needs minor revision before acceptance for publication.

  1. As mentioned above, the description, uses and conservation methods of local chicken breeds should be the focus of this manuscript. But section 6 (line 316) and section 7 (line 340) take up too much space in the whole article, and unfortunately, these two sections cannot appropriately support the conclusion “strategies to preserve and sustain them 37 must be intensified” in line 37.
  2. In line 82-86, the author mentioned that researchers have invested enormously in addressing the challenges related to the nutritional status and genetic material of local chickens, listing Badubi’s research for the corresponding, but body size information is not directly related to nutrition or gene levels. Suggest to modify the argument.
  3. In Simple Summary, please fill in the missing punctuation in line 16.
  4. Check for errors in line 253 “village chickens,.”, line 260 “breeds be preserved”, line 386 “heavy eggs four”.
  5. In line 350, the comma is in the wrong place.

Author Response

Dear Reviewer,

Authors would like to thank and acknowledge comments and suggestions made.

Regards,

Reviewer 4 Report

The Authors went through a great efforts reviewing up to date the literature regarding the Local Chicken Breeds of Southern Africa. However, a summary table of phenotypic characteristics, production/performance traits of local/ cross breed of chickens and their distribution across the country would add much weight to the review manuscript.

Author Response

Dear Reviewer,

Authors would like to thank and acknowledge comments and suggestions made. A summary table of phenotypic characteristics, production/performance traits of local/ cross breed of chickens and their distribution has been added as suggested.

Thanks and Regards,

Reviewer 5 Report

  • The similarity index showing 28%, please reduce it.
  • Make the simple summary shorter.
  • It is preferable to use some figures.
  • Revise the manuscript for English.

Author Response

Dear Reviewer,

Authors would like to thank and acknowledge the comments and suggestions made.

Regards,
